# New Mediators in the Crosstalk between Different Adipose Tissues

**DOI:** 10.3390/ijms25094659

**Published:** 2024-04-25

**Authors:** Almudena Gómez-Hernández, Natalia de las Heras, Beatriz G. Gálvez, Tamara Fernández-Marcelo, Elisa Fernández-Millán, Óscar Escribano

**Affiliations:** 1Departamento de Bioquímica y Biología Molecular, Facultad de Farmacia, Universidad Complutense de Madrid, Plaza Ramón y Cajal, s/n, 28040 Madrid, Spain; algomezh@ucm.es (A.G.-H.); bggalvez@ucm.es (B.G.G.); tafernan@ucm.es (T.F.-M.); elfernan@ucm.es (E.F.-M.); 2Departamento de Fisiología, Facultad de Medicina, Universidad Complutense de Madrid, Plaza Ramón y Cajal, s/n, 28040 Madrid, Spain; nlashera@ucm.es; 3CIBER de Diabetes y Enfermedades Metabólicas Asociadas (CIBERDEM), Instituto de Salud Carlos III (ISCIII), 28029 Madrid, Spain

**Keywords:** adipose tissue, metabolism, obesity

## Abstract

Adipose tissue is a multifunctional organ that regulates many physiological processes such as energy homeostasis, nutrition, the regulation of insulin sensitivity, body temperature, and immune response. In this review, we highlight the relevance of the different mediators that control adipose tissue activity through a systematic review of the main players present in white and brown adipose tissues. Among them, inflammatory mediators secreted by the adipose tissue, such as classical adipokines and more recent ones, elements of the immune system infiltrated into the adipose tissue (certain cell types and interleukins), as well as the role of intestinal microbiota and derived metabolites, have been reviewed. Furthermore, anti-obesity mediators that promote the activation of beige adipose tissue, e.g., myokines, thyroid hormones, amino acids, and both long and micro RNAs, are exhaustively examined. Finally, we also analyze therapeutic strategies based on those mediators that have been described to date. In conclusion, novel regulators of obesity, such as microRNAs or microbiota, are being characterized and are promising tools to treat obesity in the future.

## 1. Introduction

Adipose tissue (AT) has traditionally been considered a mere fat depot, but in reality, it is a complex and dynamic organ involved in the regulation of multiple physiological processes, including energy homeostasis, nutrition, insulin sensitization regulation, body temperature, and immune response [1]. AT is composed of different cellular types including endothelial cells, immune cells, mature adipocytes, and their progenitors. Different subtypes of adipocytes have been identified in mammals, but three main types are primarily recognized, having been named the three types of AT: white, beige, and brown adipocytes [2,3].

White adipocytes typically exhibit a unilocular structure, characterized by a single large lipid droplet occupying most of the cell volume. In the human body, WAT is primarily located beneath the skin, concentrated in the abdominal and gluteofemoral depots, and in the abdominal cavity, forming the subcutaneous white adipose tissue (sWAT) and visceral white adipose tissue (vWAT), respectively [4]. These two locations have different metabolic and physiological implications: sWAT is related to protective effects on energy homeostasis and vWAT is associated with a higher risk of metabolic disorders [5]. The main function of WAT is to store and release energy by storing triglycerides (TGs) and releasing fatty acids (FAs) for energy synthesis in response to changes in systemic energy levels (Figure 1). However, an imbalance in these processes caused by an excessive energy intake that exceeds the individual’s energy expenditure leads to the expansion of WAT through two strategies: hyperplasia and hypertrophy of adipocytes. An excessive accumulation of TG in the WAT, and consequently in other tissues, leads to a pathological metabolic state known as obesity. This disease is considered a pandemic by the World Health Organization. Particularly in Europe, the prevalence of obesity in men ranges from 4.0% to 28.3%, and in women from 6.2% to 36.5%. Obesity and its comorbidities, e.g., type 2 diabetes (T2D), cardiovascular disease, hypertension, liver dysfunction, and cancer, among others, represent a serious public health problem. Therefore, a multidisciplinary approach to the treatment of the disease, as well as the search for anti-obesity therapies, are of crucial importance [6,7,8]. WAT exerts its actions through the secretion of adipokines, cytokines, lipokines, and extracellular vesicles (EVs), among other elements. This enables it to establish a close communication with crucial organs such as the pancreas, heart, or liver [9].

Brown adipose tissue (BAT) is composed of the so-called thermogenic adipocytes, and its main function in mammals is thermoregulation. The adipocytes that make up this tissue have a high density of mitochondria, and multiple lipid droplets (Figure 1). BAT is found in cervical, supraclavicular, axillary, paravertebral, mediastinal, and abdominal regions in adult humans [10,11]. The brown adipocytes are characterized by expressing the mitochondrial uncoupling protein 1 (UCP-1). This protein is in the inner side of the mitochondrial membrane and is involved in the physiological response to cold to generate heat: once activated by FAs or β-adrenergic receptor (β-AR) agonists, it uncouples the electron transport chain inducing the release of energy as heat. Moreover, different thermogenic mechanisms independent of UCP-1 have also been described [11]. Furthermore, the existence of molecular and metabolic adaptation strategies to cold in white adipocytes has been demonstrated [12]. Brown adipocytes preferentially utilize glucose and FAs as energy substrates for thermogenesis. Specifically, exposure to cold increases glucose uptake, which is metabolized through glycolysis and mitochondrial oxidation to produce energy. Additionally, glucose is an efficient carbon source for glycerol and acetyl-CoA production, both essential in lipogenesis. Regarding the use of lipids by BAT, it is capable of taking up extracellular lipids as well as metabolizing intracellular TG through lipolysis. Other metabolites can act as energy substrates in BAT, for example, succinate, lactate, and branched-chain amino acids [13]. BAT acts as an endocrine organ through the secretion of different factors, called “batokines”, implicated in the regulation of metabolic homeostasis [14]; this point is discussed in detail in Section 3.2 of this review. Taking into account the described functions of BAT, several therapeutic strategies have been proposed, e.g., increasing thermogenic activity may be useful for treating obesity and T2D, and synthetic analogs of certain batokines have been shown to provide metabolic benefits in overweight humans [14,15].

Beige adipose tissue (BeAT) is composed of adipocytes similar to brown adipocytes, which are characterized by multilocular lipid droplets and abundant mitochondria expressing UCP-1 (Figure 1). Beige adipocytes predominantly reside in sWAT depots, located in humans at the cranial, facial, abdominal, femoral, and gluteal regions [1,16]. These cells are generated in WAT in response to certain stimuli, such as cold exposure, and can come from progenitors (de novo beige adipogenesis) [17] or from the transdifferentiation of existing white adipocytes in a reversible process known as WAT “browning” [18]. These beige adipocytes are capable of absorbing circulating glucose and lipids, increasing energy expenditure and thermogenesis. Therefore, the activation of this special adipose tissue represents an attractive therapeutic target for treating obesity and T2D [19].

The aim of this review is to broadly present an overview of novel mediators secreted by the adipose tissue, as well as others of different nature that directly impact it and promote the development of obesity. We also thoroughly cover numerous mediators that act on BAT, or are secreted by it, exerting actions on other organs with anti-obesity properties.

## 2. Mediators Triggering Obesity

Due to its multifactorial nature, obesity requires the analysis of multiple mediators across different tissues. This section attempts to describe the most relevant mediators associated with increased obesity development, which may represent potential targets for the treatment of obesity.

Obesity can be regulated through multiple interactions between endocrine tissues and the nervous system. We are now aware of the importance of AT in metabolic homeostasis and of the fact that its accumulation leads to chronic inflammation [20]. WAT is one of the main regulators of metabolism, controlling energy storage, inflammation, and immunity. The relationship between chronic inflammation and obesity is not completely understood, and several mechanisms have been discussed previously [21]. In fact, it has been demonstrated that WAT can secrete approximately more than fifty different signaling molecules and hormones, namely adipokines [22], which play a relevant role in the metabolism of glucose and immunity. Changes in its secretion profile may contribute to insulin and leptin resistance and induce the development of obesity and T2D [23,24], whereas it has been observed that the adipose tissue of lean individuals predominantly secretes anti-inflammatory adipokines such as transforming growth factor-beta (TGF β), adiponectin, apelin, IL-4, IL-10, and IL-13 [25].

Leptin is a hormone secreted by AT that regulates energy balance and appetite. It can decrease food intake, alter neuroendocrine function, and influence the metabolism of lipids and glucose. In fact, it has been demonstrated that leptin resistance may conduct insulin resistance and lipid accumulation, both in obese mice and humans. Adiponectin can be detected in low amounts in serum and is another hormone mainly secreted by adipocytes. It has been also shown that adiponectin may have a positive impact on insulin sensitivity [26]. In fact, impaired adiponectin/leptin levels produced insulin resistance in obese rats [27], and in adiponectin-deficient transgenic mice [28], improved insulin sensitivity was observed. In fact, a study comparing morbidly obese subjects with non-morbidly obese subjects showed that only with the second group a real correlation between leptin and adiponectin with different measurements of body composition could be found [29].

Cytokines secreted by WAT, such as TNF-α, IL-6, IL-1β, IL-8, or IL-18, are implicated in inflammatory processes [25] and disrupted adipokine production and secretion are strongly associated with obesity-related comorbidities [24]. In this sense, resistin seems to be involved in insulin resistance and inflammation in humans and murine models [30]. Resistin has been demonstrated to be an antagonist to insulin both in vivo and in vitro conditions [31]. In fact, resistin levels were increased in diabetic and obese mice. The exogenous administration of resistin increases glucose production and plasma levels in mice models [30]. Moreover, resistin is induced by TNF-α and has been shown to also play a role in inflammation by stimulating the production of IL-6 in humans [31].

Visfatin, a novel cytokine originating from AT, was highly expressed in visceral regions. Major human and mouse WAT accumulation positively correlated with visfatin expression in mature adipocytes. In fact, visfatin mimicked insulin signaling mainly in skeletal muscle and the liver, as observed in cultured cells and mouse models [32]. In humans, visfatin has been demonstrated to participate in the early inflammation processes of obesity and its levels are positively correlated with obesity and overweight patients with metabolic syndrome diagnosis [33].

Apelin is another recent adipokine secreted by adipocytes and distributed all over the body. Specifically, it is a regulatory peptide that has been described to be implicated in energy metabolism and physiological homeostasis after binding to its main ligand APJ (a G-protein-coupled-receptor), both in rodents and humans [34]. In fact, elevated concentrations of apelin correlate with metabolic alterations in lipid and glucose metabolism of mouse and human adipocytes, degenerating in obesity and diabetes [35,36].

Several other recent adipokines have been recently discovered and their role in obesity and metabolic pathologies are being investigated. Chemerin, increased in inflammatory conditions, has been implicated in the development of obesity and its complications [37]. In fact, chemerin-deficient mice develop an impaired glucose metabolism in skeletal muscle and the liver, mainly due to a reduction in the AKT phosphorylation pathway [38]. DPP4, a protease secreted by visceral adipose tissue, is also a proinflammatory molecule elevated in obese patients. It can affect both insulin signaling as well as macrophage infiltration [39]. In fact, hepatocyte DPP4 has been demonstrated to induce insulin resistance and activate adipose tissue macrophages in obese mice [40]. On the other hand, isthmin-1 and omentin have been described as anti-inflammatory adipokines secreted by the adipose tissue. Their levels are decreased in obese patients and in general, are correlated with an impaired glucose uptake [41]. Nevertheless, isthmin-1 gene expression levels were recently demonstrated to be correlated with the abdominal fat localization in humans, being a valuable biomarker for the detection of major obesity risks [42]. On the other hand, plasma omentin-1 levels have been demonstrated to be decreased in overweight and obese patients, recovering normal levels after weight loss. In fact, omentin-1 can induce glucose uptake via Akt activation in human adipocytes [43].

Multiple adipokines have been described to play a role in the resistance to insulin and in the maintenance of the chronic inflammation detected in obesity patients and could be classified as inflammatory mediators as mentioned previously [44]. Nevertheless, lipids, including non-esterified FAs (NEFA), are also increased in obese subjects and participate in triggering inflammation and causing lipotoxicity in different tissues [45]. FAs also played an important role in regulating adipokines secretion and their function [46].

### 2.1. Cellular Mediators Involved in Obesity

Cellular immune mediators have also been described recently and linked to the development of chronic inflammation and obesity [1]. In fact, it has been described that obesity reduces the immune system by decreasing lymphocytes and modifying monocyte and neutrophil levels. In fact, inflammation and insulin resistance in obese subjects have been demonstrated to be triggered by an impaired immune response. As the severity of obesity increases, the size of macrophage aggregates also increases, similarly to other inflammatory diseases. In obese subjects, proinflammatory cytokines produced by macrophages and T-cells were found in abundance, while in the adipose tissue of lean subjects, anti-inflammatory cytokines, M2 macrophages, and Th2 cells were mainly detected [47].

Macrophages play a fundamental role in the adipose tissue of humans and in fact are the most expressed population of the immune system. M1 macrophages have been described to trigger adipose tissue inflammation as well as insulin resistance in obesity [48,49]. However, inflammation in AT can also be mediated by other immune populations such as neutrophils or B and T-cells [50]. In fact, B-cells may produce autoreactive antibodies that increase the inflammatory state while Th1 cells produce inflammatory cytokines such as IL-6 [50].

Immunosuppressive regulatory T cells (Tregs) decline in obesity, but their regulation still requires further investigation. This decline produces major changes in the level of different cytokines such as IL-10, driving insulin resistance in obesity as previously described [51]. CD4+ Th17 cells have also been reported to be increased in the human visceral adipose tissue of morbidly obese. This population seems to increase the inflammation in adipose tissue, contributing to the development of obesity [52].

### 2.2. Novel Nediators

Novel obesity-related mediators include omics-based biomarkers such as microRNAs (miRNAs), the microbiome, and metabolites. Although several biological mechanisms have been described to trigger obesity, others, e.g., lipid metabolism (miR-181d), glucose metabolism (miR-378a), adipogenesis (miR-138-5p), and the regulation of cytokines (miR-122-5p), could be controlled by miRNAs as recently described [53]. Hence, elucidating the main role of miRNAs in obese subjects is necessary to develop novel tools to improve obesity [54], and this will be further discussed in the following section.

Moreover, several studies suggested recently that the intestinal microbiota may also be a principal mediator in the development of obesity. In fact, it has been demonstrated that both inflammation and AT composition may be modified by the microbiota contributing to the development of obesity. Moreover, the intestinal microbiota impacts directly the metabolism and energy balance by controlling the energy extracted from nutrients and affecting obesity directly. On the other hand, different studies have demonstrated that the eating habits may modify the composition of the intestinal microbiota, impacting obesity directly [55]. Beneficial bacteria (*Bacteroidetes*) can be reduced by fat and carbohydrate-enriched diets, while a similar diet tends to increase pathogenic bacteria (*Firmicutes*). The imbalanced equilibrium between beneficial and pathogenic bacteria triggers obesity and diabetes due to the induction of an inflammatory response, hormone dysfunction, and dyslipidemia [55]. Several mechanisms have been reported to link gut microbiota with obesity, highlighting among others, an increased intestinal permeability that induces dysbiosis [56] and modification of expression of host metabolism-related genes by gut microbiota [57].

Gut microbiota also produces metabolites that change during obesity development and metabolic disorders. Butyrate, propionate, indole derivatives, and polyamines, among others, are examples of metabolites produced by intestinal bacteria after processing nutrients that can mediate inflammatory responses and metabolic disbalances [58]. The short-chain fatty acids regulate different signaling pathways including peptides that control insulin resistance or inhibit appetite [59]. Moreover, acetate or propionate have been shown to impact the nervous system directly, promoting obesity and its complications [60].

More recently, it has been shown that fasting-induced adipocyte factor (FIAF) may be regulated by bacteria fermentation products and behave as an intestinal microbiota modulator, inducing obesity through affecting lipid metabolism [61].

In conclusion of this section, the current literature indicates that obesity results in the dysregulation of numerous body systems, including metabolic function, the immune/inflammatory system, and the gut microbiota. Despite extensive research in this field, no concrete mediators’ profiles have been found to correctly stratify the individuals at risk of obesity-related diseases and therefore, targeted preventive or therapeutic intervention still is controversial. The use of an integrated multi-organs analysis may advance obesity research by overcoming the challenges faced when analyzing the complex network associated with obesity-related disease (Figure 2).

## 3. Anti-Obesity Mediators

Between 2007 [62] and 2009 [63,64,65,66], the presence of active BAT deposits in human adults was demonstrated and their metabolic relevance in human physiology was described. BAT activity decreased with age [67,68,69] and is inversely correlated with BMI [70,71] and visceral adiposity [72,73].

In obesity, it has been described that BAT whitening happens, consisting of several changes in the morphology of the brown adipocyte as well as an altered gene expression profile. Alterations occur in the morphology of the tissue, where whitened brown adipocytes can be observed surrounded by a significant number of collagen fibrils. In addition, these adipocytes present an enlarged endoplasmic reticulum, cholesterol crystals, and some altered mitochondria. More importantly, the gene expression profile of whitening BAT could involve the upregulation of markers of activated inflammasome and ER stress, and the downregulation of markers of vascularization, electron transport chain, β-adrenergic signaling, and specific membrane receptors [74]. In this process of BAT whitening, associated with obesity, there are many molecular mechanisms implicated. First, the acetylation of peroxysome proliferator-activated receptor (PPAR)γ (Lys 293) present in obesity and associated with aging, decreases UCP-1 mediated by the increase in adipisin and favors the whitening of BAT [75]. On the other hand, the presence of pro-inflammatory and senescent S100A8+ immune cells derived from bone marrow, primarily T cells and neutrophils, invade BAT in subjects with obesity during aging, compromising axonal networks and thermogenic BAT function [76]. Moreover, in obesity, the activation of hypoxia inducible factor 1α (HIF-1α), which inhibits BAT thermogenesis and cellular respiration and promotes weight gain, is added to mTOR inhibition of peroxisome proliferator-activated receptor gamma coactivator-1-α (PGC-1α) [77]. Finally, the increase in brown adipocytes’ apoptosis and necrosis [78] together with the exhaustion or imbalance of the gut microbiota, alters thermogenesis [79,80].

In opposition, various mechanisms of BAT activation have been described, which could constitute future anti-obesity therapeutic strategies and which would also induce the WAT browning, such as prolonged cold exposure, exercise, and following the Mediterranean diet (rich in polyunsaturated and monounsaturated fatty acids, PUFAs and MUFAs, and different molecules). The mechanisms involved in this process that improve the functionality of BAT and beige cells involve a direct or indirect, chronic or acute activation of UCP-1 and might be mediated by the activation of coactivators of UCP-1 transcription, such as PGC1-α, β-cell factor 2 (EBF2), and PR domain zinc finger protein 16 (PRDM16) [81,82]. Moreover, other mechanisms could involve the activation of CaMKII or sirtuin 1 (SIRT1), resulting in an increased expression of thermogenic genes, including PRDM16 and PGC-1α and triggering UCP-1-mediated thermogenesis in brown adipocytes [82,83]. Another indirect mechanism of UCP-1 activation is induced by AMPK activation, favoring thermogenic BAT function by the production of free FAs or SIRT1 activation [82,84,85]. It is important to keep in mind that one of the classical mechanisms of BAT activation is mediated by β-AR, activating UCP-1 directly as targets of the CNS for fat burning or indirectly through cyclic AMP (cAMP), resulting in FA generation [86,87]. In humans, β2-AR is the predominant receptor for BAT lipolysis. On the other hand, the signaling through estrogen receptors also increased UCP-1 gene expression [88]. Finally, in beige cells, the significantly decreased ATP production by the inhibition of p38 and p-JNK signaling can upregulate UCP-1 gene expression [89].

BAT is one of the tissues that contributes to energy expenditure through thermogenesis, and it has been highlighted in recent decades how the loss of mass or function of this tissue could contribute to the development of obesity. As we had previously described for WAT, both BAT and beige cells communicate with the brain and peripheral organs through a wide variety of secretions and absorption processes—controlling adipokines, miRNAs, EVs, and metabolites—making them future candidates for the treatment of obesity and associated metabolic diseases.

### 3.1. Anti-Obesity Mediators Secreted by Other Tissues with Effect in BAT and Beige Cells

In this context, we have mentioned that BAT is an endocrine tissue, and it is also a receiver of molecules that are secreted by other tissues, such as WAT, liver, brain, or skeletal muscle that might activate BAT, which, in turn, may induce WAT browning through the beige cells’ activation. In this review, we are going to focus first on molecules secreted by other tissues that have their effects on BAT and beige cells. The molecules we want to focus on are myokines, hormones, amino acids, and new mediators such as long noncoding RNAs (lncRNAs) and miRNAs.

Within the group of myokines, irisin is a 112 amino acid peptide formed from its precursor fibronectin type III domain 5 (FNDC5) and secreted mainly by skeletal muscle in response to exercise [90]. Activated via the PGC-1α pathway [91], this myokine is important in metabolic regulation, protecting against insulin resistance and cardiovascular disease. Specifically, its role in glycemic control and insulin sensitivity has been demonstrated to be through the induction of the browning of WAT, thus contributing to global energy expenditure. It has also been described that irisin could improve the lipid profile without altering adipokine levels, which may help to prevent obesity [90]. In this sense, it has been described how treatment with irisin during the differentiation of white adipocytes significantly increased the basal mitochondrial respiration rate as well as the expression of UCP-1 and PPARγ, promoting their browning and demonstrating the effectiveness of irisin in reducing body weight [92]. Similar results were obtained years later when subcutaneous white adipocytes treated with irisin significantly increased UCP-1 expression [93]. These effects seem to be mediated by AMPK signaling since si-AMPKα1-injected mice showed partial inhibition of irisin-induced browning of WAT by down-regulating the expression of UCP-1 [93]. In contrast to these studies, preadipocytes incubated with irisin did not differentiate to brite human adipocytes and showed no changes in UCP-1 expression [94].

The thyroid hormone triiodothyronine (T3) induces thermogenesis through sympathetic innervation via a synergistic effect on β-AR, mainly β1- and β3-subtypes [95]. Stimulation of β3-AR induces PKA activation, which targets the transcription factor cAMP response element binding protein (CREB), stimulating expression of the *Ucp1* gene. In addition, the activation of PKA by the β3-AR can lead to the phosphorylation of ATGL, which induces the hydrolysis of triacylglycerols and subsequent lipolysis, thereby transforming the unilocular adipocytes into multilocular cells. T3 can also directly stimulate the coactivator PGC1-α, which interacts with PPARα/γ transcription factors [95], thereby inducing the expression of genes involved in FA oxidation, mitochondrial respiration, and biogenesis [96], as well as UCP-1 levels mediated by irisin myokine secretion, as mentioned above [90,91]. In this sense, hyperthyroid mice showed greater β-oxidation and a decrease in amino acid levels, by increasing short- and long-chain acylcarnitines in BAT [90], whereas hypothyroidism caused impaired BAT thermogenesis [97]. It is worth noting that it has also been suggested that the T3 effects on BAT thermogenesis may be regulated in a UCP-1-independent manner since the T3 treatment of thyroidectomized rats did not recover UCP-1 levels but conserved a norepinephrine-induced thermoregulatory response of BAT [98]. In this way, autophagy plays a critical role in mitochondrial turnover in BAT, but there are controversial data in the literature regarding the effect of T3 on autophagy. Yau et al. [96] reported that an induction of thermogenesis in vivo by T3 required BAT-specific autophagy activation, an effect associated with SIRT1 activation which in turn inhibited mTOR. However, others have shown that T3 acute administration to differentiated brown adipocytes induced mTOR activation and suppressed mitochondrial autophagic degradation [99]. On the other hand, T3 may be helpful in increasing leptin and adiponectin levels and in this way reduce insulin resistance [90]. Therefore, the stimulation of BAT activity by T3 seems to represent a potential therapeutic strategy for obesity.

Other mediators that might be used by BAT and have beneficial effects on energy expenditure would be branched-chain amino acids (BCAAs) and it has been described that BCAA supplementation is often beneficial for energy expenditure. Therefore, leucine and isoleucine have very similar effects in improving insulin sensitivity, reducing lipid depots, and promoting WAT browning [100]. On the other hand, an increase in circulating levels of BCAAs has been observed in obesity and diabetes [101]. Through cold exposure, BAT utilizes BCAA in mitochondria for thermogenesis and promotes systemic clearance of BCAAs in experimental models and humans [102]. Interestingly, an administration of Tirzepatide, a dual GIP and GLP-1 receptor agonist, to HFD-fed mice clearly increased the catabolism of BCAAs in BAT, suggesting a new mechanism by which this revolutionary treatment can account for significant weight loss in obese patients [103].

Regarding new mediators, long noncoding RNAs (lncRNAs) and miRNAs have been shown to play a role in brown adipogenesis, BAT thermogenesis, and the promotion of white fat browning, thereby increasing energy expenditure and decreasing body weight gain. Both lncRNAs and miRNAs may be secreted into EVs from other tissues and have their actions in BAT and beige cells. Within the group of lncRNAs, lnc266 was highlighted as promoting the browning of white fat and thermogenic gene expression in obese mice, increasing core body temperature, and reducing body weight gain. One of the mechanisms involved in these effects is that lnc266 sponged miR-16-1-3p and thus abolished the repression of miR-16-1-3p on UCP-1 expression [104]. There are many other different miRNAs also involved in the differentiation of brown adipocytes, browning, or thermogenesis (Table 1). For instance, miR-21 was found to be positively correlated with BMI [105], suggesting a causative role in adipose tissue expansion. However, miR-21 mimics up-regulated browning and thermogenesis markers *Ucp1*, *Fgf21*, *Pgc-1α*, *Tmem26*, *Sirt1*, and *Vegf-A*, both in 3T3-L1 adipocytes as well as in miR-21 treated-mice [106]. Similarly, miR-22 expression is increased in the BAT in response to cold exposure, and miR-22 deficiency, global or specific in mice AT, showed BAT whitening, reduced thermogenesis, and impaired cold tolerance [107]. In addition, miR-122-5p, a hepatic-specific miRNA, is increased in the circulating exosomes of obese patients and is negatively associated with the BAT activity [108]. In addition, miR-191-5p has been demonstrated to inhibit the conversion of WAT to BAT, targeting PRDM16. Interestingly, circulating EVs from mice subjected to long-term exercise showed reduced levels of miR-191-5p and a co-culture with 3T3-L1 fully differentiated adipocytes increased the expression of BAT markers such as *Ucp1* and *Prdm16*, while decreasing the expression of the WAT markers *Leptin* and *Adipsin* [109].

Other miRNA as miR-669a-5p plays a role in regulating adipocyte differentiation and fat browning. Its expression was increased during the adipogenic differentiation of 3T3-L1 and C3H10T1/2 adipocytes and in the iWAT of mice exposed to cold, promoting adipogenic differentiation and browning of adipocytes in vitro and in vivo [136].

Additional miRNAs whose increment could have a detrimental effect on BAT activation would be miR-27b and miR-34a, both of which are inhibitors of brown and beige adipogenesis, and their levels are decreased in response to cold exposure and β-adrenergic activation [131,132,137]. However, their levels increase significantly in obesity [132] and during differentiation [125,131,134]. miR-34a directly regulates Fgfr1 and disrupts FGF21 signaling, thereby preventing PGC1-α activation and WAT browning [132]. In this sense, let-7i-5p is also an inhibitor of thermogenesis, regulating browning marker genes such as Ucp1, Prmd16, and citrate synthase [130].

Finally, different clusters and miRNAs have been described that negatively regulate the expression of genes involved in brown adipocyte differentiation or white adipocyte browning. An miR-193b-365 cluster was described as the first miRNA that represses the myogenic potential of preadipocytes, allowing for the differentiation of brown adipocytes [134]. miR-133 directly and negatively regulates PRDM16, and the inhibition of miR-133 promotes the differentiation of BAT and WAT precursors into mature brown adipocytes [133,135]. Similarly, the decrease in miR-494-3p levels during WAT browning regulates mitochondrial biogenesis and thermogenesis through PGC1-α in beige adipocytes [135].

### 3.2. Anti-Obesity Mediators Secreted by BAT and Beige Cells with Effects in Other Tissues

In addition to the capacity of BAT to protect against chronic metabolic disease due to its ability to use glucose and lipids for thermogenesis, BAT also has a secretory role, which could contribute to the systemic effects of BAT activity. BAT and beige cells also act as endocrine tissues, secreting diverse adipokines (batokines), such as: cytokines, factors, proteins, metabokines, and EVs, which could signal and mediate different metabolic effects in target organs and contribute to an improvement in the obesity (Figure 3) [123].

The first identified adipokine secreted by BAT under conditions of thermogenic activation was fibroblast growth factor 21 (FGF21). The heart, WAT, brain, and pancreas are potential target tissues and are sensitive to FGF21 secreted from the BAT. In the heart, the cardioprotective role of FGF21 has been recognized [122]. Another recently discovered batokine is myostatin, which targets skeletal muscle and experimentally activated BAT by cold exposure, reducing myostatin levels and increasing exercise performance [129].

Other batokines released by brown adipocytes that target sympathetic nerve endings would be neuregulin-4 (NRG4), S100b protein, and nerve growth factor (NFG). NRG4 improves metabolic dysregulation in insulin resistance, obesity, NAFLD, and diabetes through the activation of various mechanisms such as anti-inflammation, the regulation of autophagy, pro-angiogenesis, and normalization of lipid metabolism [50]. The S100b protein induced neurite outgrowth from sympathetic neurons in adipose depots [77]. However, the BAT production of NGF was higher in genetically obese rats and mice and prolonged cold exposure decreased the BAT NGF synthesis in obese mice [138]. Bone morphogenetic protein-8b (BMP8B), also released by BAT, targets WAT and vascular cells. Both nutritional and thermogenic factors can induce the production of BMP8B in mature BAT and its response increases norepinephrine through the activation of p38MAPK/CREB and an increase in lipase activity [118].

In addition, the novel batokine ependymin-related protein 1 (EPDR1) has been detected in human plasma and implicated in the regulating mitochondrial respiration linked to BAT thermogenesis [120]. Moreover, EPDR1 is proposed to be a key regulator to maintain glucose homeostasis in obese people, due to the upregulation of EPDR1 that could improve β-cell function by promoting glycolysis and the TCA cycle [120,121]. In addition, active BAT is a source of C-X-C motif chemokine ligand-14 (CXCL14), which concertedly promotes adaptive thermogenesis via M2 macrophage recruitment, BAT activation, and the browning of WAT [119].

Regarding the metabokines secreted by BAT, we could highlight several metabolites that target the liver and WAT. The metabolite 3-aminoisobutyric acid (BAIBA), generated by the breakdown of thymine, is inversely correlated with cardiometabolic risk factors. BAIBA induces WAT browning and improves hepatic β-oxidation [110]. Other small metabokines are 3-methyl-2-oxovaleric acid (MOVA), 5-oxoproline (5-OP), and β-hydroxyisobutyric acid (BHIBA), which are synthesized by BAT and beige cells and are secreted by monocarboxylate transporters. They are associated and correlated with browning markers of WAT and inversely associated with body mass index [124].

During thermogenesis induced by cold exposure, it has been described that the secretion of human BAT EVs (BDEVs) increases significantly, suggesting their possible participation in this activity [139]. Similarly, beige adipocytes release more EVs when activated by cAMP signaling and contain factors that could have a protective effect for diabetes [138]. So, BAT-derived miR-378a-3p in BDEVs reprogramed systemic glucose metabolism by inducing hepatic gluconeogenesis targeting p110α during cold stress [114]. Moreover, when BAT was activated by cold exposure, miR-132-3p was also increased in BDEVs attenuating Srebf1 hepatic expression, and in consequence it regulates hepatic lipogenesis [115]. In addition to cold exposure, exercise also increases the secretion of EVs by BAT that had a cardioprotective effect with the release in the heart of cardioprotective miRNAs, such as: miR-125b-5p, miR-128-3p, and miR-30d-5p [126]. The other miRNAs produced by BAT, miR-26 and miR-32, are also involved as positive regulators of brown adipogenesis, BAT thermogenesis, and inguinal WAT browning [125,127]. In addition, miR-32 increases Tob1, and both appear to be modulators of FGF-21 signaling [127], and miR-26 targets metallopeptidase domain 17 (ADAM17), increasing UCP1 and PGC1-α [125]. Finally, it has been demonstrated that BAT transplantation partially improved liver lipid metabolism, oxidative stress, and fibrosis in diabetic mice through increasing circulating miR-99a, which targets NOX4 in the liver [128].

## 4. Therapeutic Strategies for Obesity

Along with cold, exercise and diet are the physiological triggers known to activate the thermogenic activity of BAT [140,141] (Figure 3). Healthy subjects under controlled cold exposure conditions showed an increase in FA uptake by BAT and oxidative metabolism by up to 182%. Likewise, BAT volume increased by 45% in the 3 h after exposure to cold, indicating a rapid reduction in TG content [140]. These data indicate that cold exposure is a powerful stimulus that targets insulin sensitivity and lipid metabolism to promote WAT browning. Accordingly, tissue-specific loss of p85α in BAT is able to prevent HFD-induced obesity in mice by promoting the activation of BAT, the amelioration of the proinflammatory profile, and consequently leading to the improvement of systemic insulin sensitivity due in part to increased IRB/IRS-1 association and insulin signaling, as well as decreased JNK activation [142]. On the contrary, BAT insulin receptor knockout mice show severe brown lipoatrophy and a strong susceptibility to obesity together with metabolic alterations [72,143]. Recently, UCP-1-activated thermogenesis has received much attention among various AT proteins as a potential target for the treatment of metabolic diseases such as obesity.

One of the new possible therapeutic targets described in metabolic diseases is 12,13-dihydroxy-9Z-octadecenoic acid (12,13-diHOME), an oxylipin, which is a bioactive metabolite released after the oxidation of PUFA, a product of linoleic FA metabolism provided in the diet [144,145]. Its blood levels increase after exposure to cold in both humans and mice [144,146]. In humans, plasma levels of 12,13-diHOME are negatively correlated with BMI, insulin resistance (HOMA-IR), and the expression of enzymes involved in protein synthesis has been shown to be selectively regulated in the BAT of mice after exposure to low temperatures [147]. The molecular mechanism indicates that this oxylipin increases FA absorption by inducing the translocation of FA transporters such as FA transport protein 1 (FATP1) and CD36 to the cell membrane increasing thermogenesis [146]. In a human lipidomic analysis, the authors demonstrated that 12,13-diHOME was a circulating lipokine released during moderate-intensity training independent of age, sex, activity levels, BMI, or fat mass [148]. An original work demonstrated that prolonged high-intensity cycling exercise in male athletes elevated serum 12,13-diHOME levels and other linoleic acid metabolites such as 9,10-diHOME [149]. Additionally in humans, after a cardiopulmonary exercise test an increase in plasma 12,13-diHOME of 23% was obtained [150], whereas in human patients with heart disease it was decreased [151]. On the other hand, works performed in mouse models showed that BAT was the main source of exercise-induced circulating 12,13-diHOME, increasing FA uptake and oxidation by skeletal muscle [147]. These results propose 12,13-diHOME as a lipokine with a promising potential for the treatment of metabolic disorders. However, many more studies would be necessary to truly understand its mechanism of action.

BAIBA, a muscle-derived exercise mimetic, has been reported to participate in lipid regulation, promoting white fat browning and increasing FFA acid oxidation via the AMPK and PPARδ signaling pathway. In addition, it decreases blood glucose, attenuates insulin resistance [110,111,152], and promotes osteogenic differentiation [112]. One of these recent studies in which BAIBA is related to lower postprandial glucose concentrations in adults with obesity, suggests that adiponectin would modulate this effect by favoring the expression of the insulin gene or the exocytosis of insulin granules [111]. On the other hand, in human aortic or umbilical vein endothelial cells BAIBA treatment induced a downregulation of proinflammatory genes and enhanced the expression of antioxidants and molecules related to mitochondrial biogenesis through an enhanced expression of PGC-1β [113]. Studies have not observed side effects, suggesting that BAIBA could be used for cardiovascular and metabolic protection, considering that it confers the benefits of exercise, especially in older sedentary subjects.

There are currently numerous molecules of synthetic and natural origin under study (Figure 3). For example, a phosphodiesterase inhibitor such as sildenafil increased UCP-1 expression approximately 4.6-fold and improved thermogenesis in overweight adults [153]. β1-AR stimulation with isoproterenol and dobutamine activates UCP-1 through cyclic AMP, resulting in generation of FAs in a human BAT cell model and fresh human BAT biopsies [154]. However, in another study, the predominant receptor for BAT lipolysis and UCP-1-mediated thermogenesis was shown to be the β2-AR (formoterol) through its chronic activation in human BAT from the deep neck region rather than β3-AR signaling pathway clearly established in rodents [155]. In explants of human visceral (omental) and abdominal sWAT, PPARγ agonist rosiglitazone induced the transcription of regulators of brite/beige adipocytes (PGC1α, PRDM16), triglyceride synthesis (GPAT3, DGAT1), and lipolysis (ATGL). Furthermore, rosiglitazone increased the expression of genes involved in FA oxidation (UCP-1, FABP3, PLIN5 protein), FA oxidation rates, and levels of electron transport complex proteins, suggesting better respiratory capacity as confirmed in newly differentiated adipocytes [156]. Studies with glucagon-like peptide-1 (GLP-1) receptor agonists on the impact on BAT in humans are more limited [157]. In this regard, exenatide reduced body weight primarily due to a reduction in lean body mass. Using 18F-FDG-PET/CT, it was noted that exenatide increased the metabolic volume and mean standardized uptake value of cervical and supraclavicular BAT depots, as well as superior mediastinal, axillary, and paravertebral BAT depots in men [158]. In animal models, the effect of centrally administered proglucagon-derived peptides such as GLP-1 was explored on BAT activity. Intracerebroventricular injection of these peptides reduced body weight and increased BAT activity, related to an increased activity of sympathetic nerve fibers innervating BAT [159].

Taking into account the cardiometabolic side effects of some of these drugs, natural and herbal products are being tested as alternative activating stimuli and it is hypothesized that they are possible agents in the prevention and therapy of metabolic pathologies [160,161] (Figure 3). However, the toxicological and pharmacological properties should always be evaluated to verify the effectiveness of the herb or natural product against serious diseases and its safe consumption for patients [162]. For instance, Chrysin, a flavonoid obtained from plant extracts from *Passiflora* species, honey, and propolis, has been proposed as a potential anti-obesity drug since it inhibits pancreatic lipase in rats under a high fructose diet. This enzyme participates in the hydrolysis and digestion of fat, cholesterol esters, and fat-soluble vitamins [163]. The result of this flavonoid was to decrease body weight gain, BMI, abdominal circumference/thoracic circumference ratio, adiposity index, calorie intake while it induced an increase in fecal cholesterol, and the locomotor activity of the rats [163]. Moreover, in a recent study it has been described that the administration of Bergacyn^®^, an innovative formulation that consists of a combination of bergamot polyphenolic fractions and *Cynara cardunculus*, in mice fed a Western diet, induced a decrease in body weight and total fat mass, and an improvement of the hyperglycemia, total cholesterol, and LDL levels. The decreased WAT depots correlated to an increase in BAT mass and a downregulation of PPARγ and prevented NF-κB overexpression, improving oxidative metabolism and inflammatory status [164]. In this regard, berberine, which is commonly used to treat diarrhea, increased BAT mass and activity and induced a decrease in body weight, and improved insulin sensitivity in overweight patients with non-alcoholic fatty liver disease after 1 month of treatment [165]. Furthermore, they showed an increase in the transcription of PRDM16, a key molecule involved in brown/beige adipogenesis, associated with the activation of AMPK, contributing to elevated systemic energy expenditure. Most of the studies carried out with resveratrol, a polyphenol produced naturally in numerous plants and fruits such as peanuts, blackberries, blueberries, and, above all, in grapes and red wine, have been carried out in 3T3-L1 preadipocytes and animal models with effects on obesity, thermogenesis, and lipid oxidation, as well as the main molecular mechanisms involved [166]. Interestingly, in human Simpson–Golabi–Behmel syndrome (SGBS), where weight is gained at an unusual rate, this natural polyphenol regulated the number and function of human adipocytes in an SIRT1-dependent manner. This sirtuin was also responsible in part for the effect of resveratrol on the lower expression and secretion of IL-6 and IL-8 in those patients [167]. Through in vitro studies, resveratrol has been shown to activate AMPK and subsequently PGC-1α, PRDM16, and PPARγ, leading to a catabolism of stored TAG and generation of FA, as well as activation of thermogenesis via UCP-1 [168]. However, the exact mechanism of its action in metabolic pathologies is still unclear and needs further research. Another natural product such as capsaicin, used topically to relieve pain in pathologies such as diabetic neuropathy, showed results such as the direct activation of adipocyte browning, the expression of UCP-1, mitochondrial biogenesis, energy consumption rates, and glycerol recycling in brown adipocytes obtained from human dermal fibroblasts with chemical stimulation [169]. Most of its effects are mediated by the activation of the sympathetic nervous system through the transient receptor potential vanilloid 1 (TRPV1) [170] in sensory neurons [171,172], also promoting the release of insulin and increasing the levels of GLP-1. Studies in preadipocytes obtained from vWAT or adipocytes from sWAT have shown that incubation with genistein [173], an isoflavone derived from soy, or with xanthohumol [174], a flavonoid from the hop plant, increased the differentiation and browning of these cells. Genistein exerted positive effects on cell viability and mitochondrial membrane potential, as well as antioxidant effects; and xanthohumol modulated mitochondrial function through the expression of mitofusin 2 (Mfn2) and CIDE-A and TBX-1 as specific markers of beige adipocytes through the PGC1α and AMPK signaling pathway.

It has been shown that breastfeeding is a key factor in the modulation of adipose tissue and, therefore, in the development of overweight or obesity during childhood [175]. Breastfed infants showed elevated levels of UCP-1, and adipocytes with UCP-1-enriched mitochondria and abundant multilocular lipid droplets. This effect is due in part to breast milk alkylglycerols promoting the activation of BeAT and the number of mitochondria in inguinal AT during infancy, in addition to preventing the transdifferentiation of BeAT into WAT [176] (Figure 3). These bioactive compounds are found in the unsaponifiable material of some marine oils, so they could be supplemented in the diet during the first stages of life.

All these studies suggest that certain natural products could be a therapeutic strategy for metabolic disorders related to obesity by activating BAT. However, further clinical trials and studies of the dose range of phytochemicals and side effects are needed to expand the potential use of drugs to induce browning or increase BAT activation with the perspective of treating metabolic pathologies and associated comorbidities.

Another aspect that has become increasingly important is the modulation of BAT function by the gut microbiota. Given that diet is a key factor that alters gut microbiota, the administration of prebiotics, probiotics, and symbiotics exerts an effect on it, and could be potential agents in browning and BAT activation (Figure 3). For instance, the administration of *Clostridium butyricum* Strain CCFM1299 (a butyrate-producing microorganism) in C57BL/6J mice under HFD was able to reduce the weight gain by increasing the energy expenditure and the expression of genes involved in BAT thermogenesis like *Ucp1*, *Pparg*, *Pgc1a*, and *Prdm16* [177]. In a similar way, in specific pathogen-free (SPF) male mice fed with a high-fat diet, the use of prebiotics and symbiotics significantly reduced the body weight gain, improved the HOMA-IR, and reduced circulating insulin and cholesterol levels [178], suggesting that the restoration of certain microbial populations with novel symbiotics is a promising approach for obesity treatment. Moreover, in overweight or obese humans, SCFAs produced through the fermentation of dietary fiber and resistant starch by gut microbiota increased fasting fat oxidation and resting energy expenditure, suggesting their effects on insulin sensitivity. Furthermore, a significant positive association between PRDM16, UCP-1, and thyroxine deiodinase 2 (DIO2) was demonstrated in sWAT and the relative abundance of *Firmicutes*, which is positively associated with circulating acetate levels [80].

Numerous studies have identified AT as a potential therapeutic target in obesity and related metabolic disorders [19,179]. Although there is no approved clinical study showing a specific therapeutic effect on this tissue [180,181], there are many successful studies in preclinical models in which gene therapy approaches have been validated. The pioneer works using adeno-associated virus (rAAV) vectors with engineered serotype Rec2 were able to transduce BAT without affecting the gastrointestinal track. With these tools, it was demonstrated that overexpressing VEGF in BAT via the oral administration of Rec2-VEGF vector induced an increase on BAT mass and thermogenesis. However, the lack of VEGF in BAT disturbed cold adaptation and decreased BAT mass [182]. In this sense, in a murine model of diet-induced obesity, the improvement of the antioxidant capacity of visceral fat through lentiviral gene therapy to restore the expression of heme oxygenase-1 (HO-1) prevented an increased adipocyte cell size, fibrosis, decreased mitochondrial respiration, the induction of inflammatory adipokines, insulin resistance, vascular dysfunction, and impaired heart mitochondrial signaling [183]. These data demonstrate that the specific expression of HO-1 in adipocytes has an important impact on distal organs. Moreover, during obesity, hypertrophic enlargement of WAT promotes lipid deposition in other tissues such as liver or muscle, inducing insulin resistance. In contrast, WAT hyperplasia is associated with the maintenance of insulin response. Regarding this, a gene therapy approach in ob/ob mice to induce BMP7 overexpression in WAT was able to improve insulin sensitivity and white adipogenesis [116]. In a more recent study, the same authors have proven that the treatment of HFD-fed mice and ob/ob mice with liver-directed AAV-BMP7 vectors enabled a maintained increase in its circulating levels. The increased BMP7 concentration was able to induce WAT browning and BAT activation, enhancing the energy expenditure, and reversing WAT hypertrophy, liver steatosis, and WAT and liver inflammation, finally resulting in an improvement in body weight and insulin resistance [117]. In a similar way, in C57Bl6/J mice under a high-fat diet, Wagner, G. et al. were able to transduce epididymal WAT (eWAT) with AAVs expressing LIM domain only 3 (LMO3), improving insulin sensitivity, mitochondrial function, and healthy vWAT expansion paralleled by increased serum adiponectin [184]. To understand some of the mechanisms associated with these effects, the authors observed that the expression of LMO3 in 3T3-L1 adipocytes increased the transcriptional activity of PPARγ, insulin-stimulated GLUT4 translocation, and glucose absorption, as well as mitochondrial oxidative capacity and FA oxidation, where the PPARγ coactivator NCOA1 has a key role.

One of the newest strategies today could be to balance the metabolic functions of AT by regulating gene expression through miRNAs or lncRNAs. In relation to the biology of BAT, miR-26, let-7i, and miR-125b among others have been identified as common miRNAs between mice and humans [185,186]. It is important to consider whether the results obtained in animal models with metabolic alterations can be transferred to humans. It must be taken into account that the amount of BAT in murine models is much greater and therefore the darkening process is induced better than in humans, also considering that overweight or obese patients have less functional BAT than healthy subjects [187]. Therefore, miRNA therapy could be less effective in humans than in the preclinical studies performed to date.

In human multipotent adipose-derived stem cells, the miR-26 family, and its effector ADAM, a disintegrin and a metalloprotease domain (ADAM17) induced signaling pathways involved in energy release, modified mitochondrial morphology similar to that observed in BAT, and promoted respiration uncoupling by significantly inducing PPARγ signaling and therefore UCP-1 expression [125]. Through pegylated lipid nanoparticles complexed with miRNA (PEG-LNP), a study performed in human preadipocytes derived from the stromal cell fraction of sWAT showed that transfection with miR-26a induced a browning of white adipocytes with an increased number of mitochondria and higher UCP-1 levels. On the contrary, miR-27 inhibited adipogenesis by downregulating PPARγ and reducing the formation of lipid droplets [188]. Another miRNA that has been related to obesity is miR-21. Its levels were increased in WAT from non-diabetic obese compared to normoweight humans and mice [106]. In addition, in in vitro experiments performed in adipocytes, miR-21 mimics regulated genes involved in WAT function and promoted browning and thermogenesis. More importantly, miR-21 mimics administration in vivo blocked HFD-induced obesity by increasing WAT browning and BAT thermogenesis through VEGF-A, p53, and TGF-β1 signaling pathways. In a very recent study, the same group has demonstrated that a new delivery approach based on a novel delivery tool based on gold nanoparticles and Gemini surfactants (Au@16-ph-16) is able to work as a potential anti-obesity drug even at low doses, replicating the positive effects of miR-21 mimics on weight gain, browning, and thermogenesis [189]. Regarding lncRNAs, a gene therapy approach with an adeno-associated virus expressing lnc266 in iWAT via in situ injection was able to stimulate the thermogenic program in this tissue in a cold environment. The molecular mechanism implies that lnc266 sponges miR-16-1-3p, inhibiting its repression on *Ucp1* expression [104].

In addition to gene therapy approaches focused on the adipose organ, there have been many reported studies in which other tissues such as liver and skeletal muscle can be targeted to improve the metabolic dysfunction observed in obesity. For instance, the hepatic-specific expression of BMP4 in mature mice through AAVs, although it reduced BAT activation through the decrease in UCP-1 expression, it was able to protect against obesity by inducing the browning of sWAT through the induction of UCP-1 expression and mitochondrial biogenesis [190]. Another tissue that can be targeted as an anti-obesity approach is the skeletal muscle since this tissue secretes several myokines such as irisin and FGF21 that participate in lipid metabolism. Recently, Zhu, H. et al. have demonstrated in a mouse model of diet-induced obesity, that the CRISPRa-based activation of *Fgf21* and *Fndc5* in skeletal muscle can improve obesity by increasing the secretion of FGF21 and irisin by myocytes and the browning of white adipocytes through the induction of UCP-1 expression [191].

## 5. Conclusions

In this review, we have highlighted how the dysregulation of different mediators secreted by adipose tissue itself, as well as cells of the immune system, miRNA, or the gut microbiota, are implicated in the development of obesity. On the other hand, BAT activation and WAT browning have anti-obesity effects and different factors involved in this process have been discussed: UCP-1 activators, cold exposure, exercise, a diet enriched in MUFAs and PUFAs, an improvement in the intestinal microbiota populations, and supplements with natural or herbal products as polyphenols, among others. Finally, we have collected the most innovative treatments designed in experimental models and clinical trials that have considered the different mediators of obesity as therapeutic targets. Those approaches are mainly focused on the reduction of WAT inflammation, the improvement of BAT functionality, mass, and activity, as well as the activation of beige cells and consequently the browning of WAT.

Several limitations of the referenced studies should be noted. Most of them were carried out in animal models, which offer several advantages, such as control of the environment or the possibility of isolating the tissues or samples to be analyzed. However, the biological and physiological gap between species seems to be a relevant limitation and makes the translation of results difficult. In some cases, the reproducibility of the data obtained in human studies is quite controversial. In the future, more clinical trials should be developed to ensure the viability of the proposed therapies, and to apply the procedures in a personalized way, taking into account the gender, race, and medical comorbidities of the patients.

## Figures and Tables

**Figure 1 ijms-25-04659-f001:**
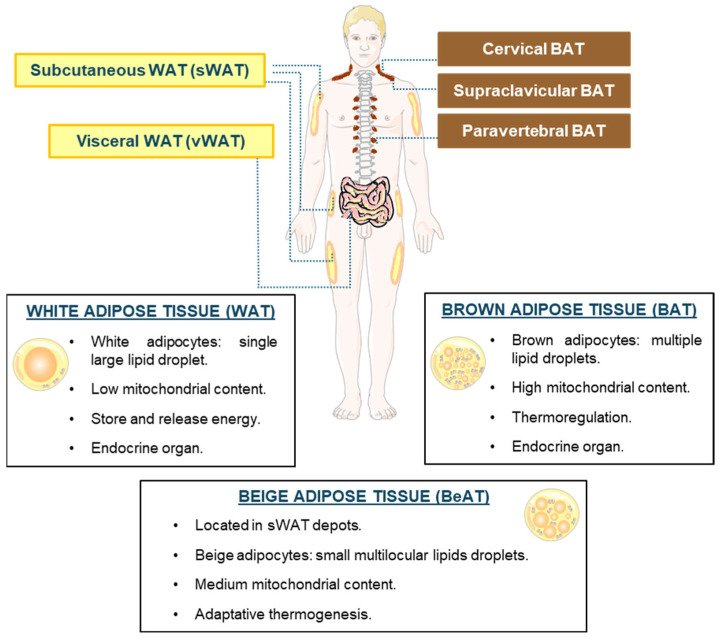
Adipose tissues: locations and functions. Illustrative representation of the main locations of adipose tissues, characteristics of adipocytes, and functions.

**Figure 2 ijms-25-04659-f002:**
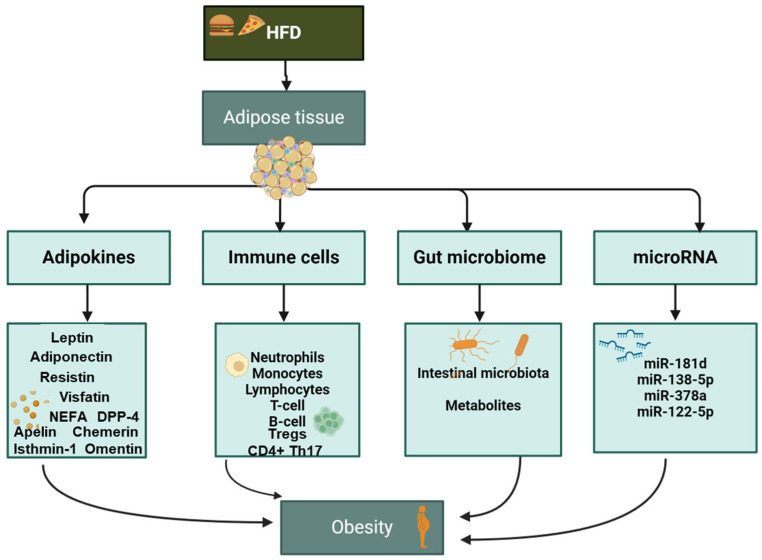
Mediators contributing to the state of chronic inflammation, leading to increased risk of obesity. Pro-inflammatory mediators as adipokines (first column), inflammatory cellular mediators as immune cells (second column), modifications in gut microbiota (third column), and changes in miRNA levels (fourth column). High fat diet (HFD).

**Figure 3 ijms-25-04659-f003:**
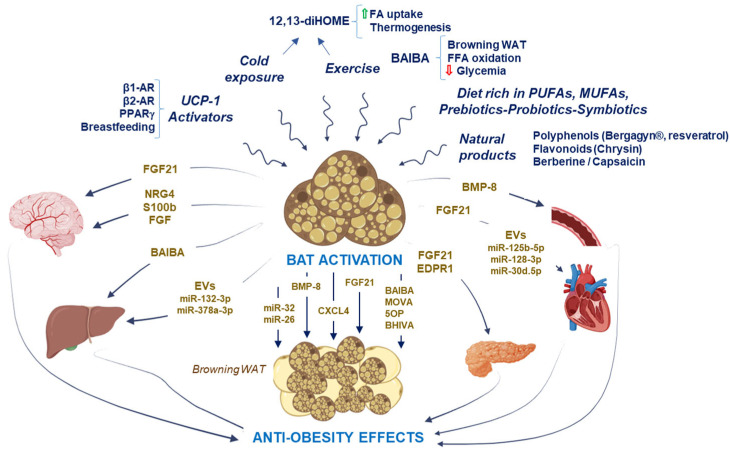
Mediators produced by BAT with anti-obesity effects. BAT activation by UCP-1 activators, cold exposure, exercise, a diet rich in MUFAs and/or PUFAs, prebiotics, probiotics, symbiotics administration, or natural products provokes the secretion of diverse adipokines (batokines), such as: cytokines, factors, proteins, metabokines, and BAT-derived miRNAs in EVs, which could signal and mediate different metabolic effects in target organs, contributing to an improvement in obesity. Green arrow shows an increase in FA uptake and red arrow represents a decrease in blood glucose levels.

**Table 1 ijms-25-04659-t001:** Anti-obesity mediators secreted by BAT, BeAT, and/or other tissues with metabolic effects.

Mediator	Metabolic Effects	Study Models	Reference
Main anti-obesity mediators secreted by BAT and BeAT with effects in other tissues			
BAIBA	Improvement of hepatic β-oxidation WAT browning Glucose metabolism Osteogenic differentiation	Humans, mice, cells	[110,111,112,113]
BDEVs	Protective effects for diabetes	Mice, cells	[114,115]
BMP7	Improvement of insulin sensitivity WAT browning BAT activation	Mice	[116,117]
BMP8B	Increase in p38MAPK, lipase activity, and sympathetic activation of BAT	Mice, cells	[118]
CXCL14	Improvement of adaptive thermogenesis M2 macrophage recruitment BAT activation WAT browning	Mice, cells	[119]
EPDR1	Improvement of β-cell function and glucose homeostasis Brown fat cell development	Humans, cells	[120,121]
FGF21	Cardioprotective	Mice	[122]
	WAT browning	Mice	[123]
MOVA/5-OP/BHIBA	WAT browning Improvement of Body Mass Index	Humans, mice, cells	[124]
miR-26	Regulates ADAM17, increasing UCP-1 and PGC1α WAT browning Brown adipogenesis BAT thermogenesis	Cells	[125]
miR-30d-5p	Cardioprotective	Mice, cells	[126]
miR-32	WAT browning Brown adipogenesis BAT thermogenesis	Mice	[127]
miR-99a	Improvement of hepatic lipid metabolism and oxidative stress Target (NOX4)	Mice	[128]
miR-125-5p	Cardioprotective	Mice, cells	[126]
miR-128-3p	Cardioprotective	Mice, cells	[126]
miR-191-5p	Inhibition of adipose differentiation, BAT activation, and WAT browning	Mice, cells	[109]
Myostatin	BAT Activation Muscle Activation	Mice, cells	[129]
NRG4	Anti-inflammatory Regulation of autophagy Energy homeostasis Glucolipid metabolism	Mice Rats Humans Cells	[50]
Main anti-obesity mediators secreted by other tissues with effects in BAT and BeAT			
BCAA	Insulin sensitivity Improvement of energy expenditure	Humans, mice	[101,102,103]
Irisin	Improvement of insulin resistance and lipid profile Glycemia control WAT browning	Humans	[90,91,92,93,94]
T3 Hormone	Increased mitochondrial biogenesis and mitophagy Improvement of insulin resistance Glycemia control WAT browning	Humans, mice	[90,95,96]
Lnc266	WAT browning (sponged miR-16-1-3p)	Mice	[104]
miR-16-1-3p	Reduced UCP-1	Mice, cells	[104]
miR-21	WAT browning Brown adipocytes differentiation Increased thermogenesis	Humans	[105,106]
Let-7i-5p	Inhibition of thermogenesis	Humans, mice	[130]
miR-22	WAT browning	Mice	[107]
miR-27b		Humans, mice	[131]
miR-34a		Mice, cells	[132]
miR-133		Mice, cells	[133]
miR-122-5p	Brown adipocyte differentiation Increased thermogenesis	Humans	[108]
miR-193b-365		Mice, cells	[134]
miR-494-3p		Mice, cells	[135]
miR-669a-5p		Mice, cells	[136]

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
