# Peer review of "New Mediators in the Crosstalk between Different Adipose Tissues"

_ijms, 2024, doi:10.3390/ijms25094659_

Round 1

Reviewer 1 Report

Comments and Suggestions for Authors

In this work, Gómez-Hernández et al conducted a review about the main players involved in the crosstalks between white and brown adipose tissues. Adipokines and cytokines secreted by adipose tissues, cellular mediators infiltrated into adipose tissues, as well as intestinal microbiota and their metabolites are reviewed in this work. In addition, mediators that promote the activation of beige adipose tissues, including  myokines, batokines, hormones secreted by other tissues, lncRNAs and miRNAs, are also examined. Finally, the authors analysed therapeutic strategies based on those mediators. Given the high prevalence of obesity and associated metabolic diseases, this review paper is timely important. It could be an important part of the literature review in the area of adipobiology.

Specific comments -

1. The review topics are too broad, and the authors could not systematically made a comprehensive review of each of those main players involved in the control of adipose tissue activity. For example, in adipokines section, the authors only mentioned briefly about leptin, adiponectin, resistin, and visfatin, without going into details. Many new and important adipokines are also missing in the review, for example, apelin, asprosin, etc. Similarly, the review about the cellular mediators infiltrated in adipose tissues, and intestinal microbiota and their metabolites are very brief without going into details. For example, the authors only added one sentence description for regulatory T cells (Tregs) which play a critical role in adipose inflammation.

2. It will be better if the authors summarize the studies about those mediators, for example, adipokines, myokines, batokines, LncRNAs, miRNAs, and immune cells, into the relevant Tables.

Comments on the Quality of English Language

NiL

Author Response

Reviewer 1

In this work, Gómez-Hernández et al conducted a review about the main players involved in the crosstalks between white and brown adipose tissues. Adipokines and cytokines secreted by adipose tissues, cellular mediators infiltrated into adipose tissues, as well as intestinal microbiota and their metabolites are reviewed in this work. In addition, mediators that promote the activation of beige adipose tissues, including myokines, batokines, hormones secreted by other tissues, lncRNAs and miRNAs, are also examined. Finally, the authors analysed therapeutic strategies based on those mediators. Given the high prevalence of obesity and associated metabolic diseases, this review paper is timely important. It could be an important part of the literature review in the area of adipobiology.

Specific comments -

  1. The review topics are too broad, and the authors could not systematically made a comprehensive review of each of those main players involved in the control of adipose tissue activity. For example, in adipokines section, the authors only mentioned briefly about leptin, adiponectin, resistin, and visfatin, without going into details. Many new and important adipokines are also missing in the review, for example, apelin, asprosin, etc. Similarly, the review about the cellular mediators infiltrated in adipose tissues, and intestinal microbiota and their metabolites are very brief without going into details. For example, the authors only added one sentence description for regulatory T cells (Tregs) which play a critical role in adipose inflammation.

We agree with the reviewer that the submitted review is quite extensive. We have tried to highlight different aspects relevant for obesity but unfortunately due to the limitation of space we could not extend as much as we would have liked in each paragraph. Nevertheless, we would like to thank the reviewer for the suggestion to enrich different parts of the manuscript.

Now we have incorporated new mediators in the adipokines section (Page-4, lines 161-174) as well as we have further explained cellular mediators or microbiota sections, including new references (Page-5, lines 200-205, 225-228 and Page-6 lines 233-239). We have not gone further in the mechanisms since the following sections and the new added table contain that information..

  1. It will be better if the authors summarize the studies about those mediators, for example, adipokines, myokines, batokines, LncRNAs, miRNAs, and immune cells, into the relevant Tables.

At the request of the reviewer, we have added a table (Table 1, Pages 9-10) summarizing the main anti-obesity mediators involved in the studies described related to adipokines, myokines, miRNAs,... as well as their most important metabolic effects.

Reviewer 2 Report

Comments and Suggestions for Authors

I must commend the authors for this systematic review of the communication between different adipose tissues and signaling molecules associated with obesity. The rich content of the paper, which covers a wide range of literature, indicates that the authors have an in-depth knowledge of the field. The following are some of my specific suggestions and comments on the paper: 

The presentation of the introduction is too detailed and could be streamlined to highlight the importance and novelty of the study.    The background and implications of the studies mentioned in the introduction can be more focused and clearly connected to the topic of the paper.

The paper uses a large body of literature to support the argument, but lacks the author's own research results. Is it possible to consider adding some of the author's own research or analytical results to enhance the persuasive power of the paper?

The discussion is exhaustive, but the logical relations between some of the sections are not clear enough.  It is recommended that the author reinforce the connections and comparisons between the various results in the discussion so that the reader can more easily understand the author's ideas. In addition, the authors should consider more fully the potential significance and limitations of individual findings and suggest some directions for future research.

The conclusions should summarize more clearly the main findings of the authors and echo the introduction. The concluding section can further highlight the importance of this topic and the potential implications for future research.

There are some grammatical and expressive errors in the paper. It is recommended that the authors examine it carefully and make revisions. Some of the technical terms in the paper can be further refined to ensure that the reader has an accurate understanding of the author's intentions.

The citation format in the paper does not appear to be uniform. Authors are advised to check and unify citation formats to improve the overall quality of papers.

Overall, this is an informative and enlightening paper. I am confident that the authors' research will be of great help to researchers in the field of obesity. With the changes suggested above, I am confident that this paper will be able to better present the progress of related research. Expect to see an improved version by the author!

Comments on the Quality of English Language

There are some grammatical and expressive errors in the paper. It is recommended that the authors examine it carefully and make revisions. Some of the technical terms in the paper can be further refined to ensure that the reader has an accurate understanding of the author's intentions.

Author Response

We want to thank the reviewer for his/her comments. We think that the manuscript is now more complete and comprehensible for the potential reader based on the comments of the three reviewers. In the responses to the reviewers, we have tried to harmonize the comments of all of them.

I must commend the authors for this systematic review of the communication between different adipose tissues and signaling molecules associated with obesity. The rich content of the paper, which covers a wide range of literature, indicates that the authors have an in-depth knowledge of the field. The following are some of my specific suggestions and comments on the paper: 

The presentation of the introduction is too detailed and could be streamlined to highlight the importance and novelty of the study. The background and implications of the studies mentioned in the introduction can be more focused and clearly connected to the topic of the paper.

As the referee suggests, we have modified the introduction trying to focus the aim of the review. (Page 3, lines 111-115).

The paper uses a large body of literature to support the argument, but lacks the author's own research results. Is it possible to consider adding some of the author's own research or analytical results to enhance the persuasive power of the paper?

We have included some data regarding our results considering the impact of some mediators or signalling pathways in BAT activity and browning (Page 13, lines 493-498). We have also included ref 76 (results from the authors) regarding the role of exercise in BAT activation and WAT browning.

The discussion is exhaustive, but the logical relations between some of the sections are not clear enough.  It is recommended that the author reinforce the connections and comparisons between the various results in the discussion so that the reader can more easily understand the author's ideas. In addition, the authors should consider more fully the potential significance and limitations of individual findings and suggest some directions for future research.

We thank the reviewer for the comment. We have already introduced a new paragraph at the end of the conclusion section where we have analysed the significance and limitations of the studies, and we have suggested several future research lines to improve the research in the obesity field (Page 17, lines 719-727).

The conclusions should summarize more clearly the main findings of the authors and echo the introduction. The concluding section can further highlight the importance of this topic and the potential implications for future research.

We have already modified the conclusions (Page 17, lines 707-713),

There are some grammatical and expressive errors in the paper. It is recommended that the authors examine it carefully and make revisions. Some of the technical terms in the paper can be further refined to ensure that the reader has an accurate understanding of the author's intentions.

We have made a complete revision of the grammar and technical terms throughout the manuscript, trying to improve the reader's understanding.

The citation format in the paper does not appear to be uniform. Authors are advised to check and unify citation formats to improve the overall quality of papers.

To manage bibliographic citations, we have used the Mendeley – Reference Management Software, which inserts citations into the text and organizes references at the end of the manuscript automatically. We have revised everything so that the format is as requested by the journal.

Overall, this is an informative and enlightening paper. I am confident that the authors' research will be of great help to researchers in the field of obesity. With the changes suggested above, I am confident that this paper will be able to better present the progress of related research. Expect to see an improved version by the author!

Comments on the Quality of English Language

There are some grammatical and expressive errors in the paper. It is recommended that the authors examine it carefully and make revisions. Some of the technical terms in the paper can be further refined to ensure that the reader has an accurate understanding of the author's intentions.

As suggested by the referee we have performed an extensive revision of the English language in order to solve that issue.

Reviewer 3 Report

Comments and Suggestions for Authors

Dear Author

The review article “New mediators in the crosstalk between different adipose tissues”. The topic is interesting, sounds well to readers, and is suitable for the journal.

Well, the manuscript is good, and the author, please incorporates my suggestion to enhance the effectiveness of this manuscript for the esteemed journal and scientific world. All comments are shared below:

Comment 1: Kindly look about the Grammer mistake in full manuscript.

Comment 2: Author adds few lines regarding peoples have issues or problem from adipose tissue

Comment 3: Add the table regarding the in vitro and in vivo studies related to adipose tissue

Comment 4: Kindly strong the mechanism part of anti-obesity mediators secreted by other tissues with effect in BAT and beige cells

Comment 5: Author also mentions about herbal approach to cure adiposity or obesity

Comment 6: Kindly rewrite the conclusion part

Comment 7: Author can use these papers to justify the herbal part and mechanism part regarding obesity.

ttps://doi.org/10.3390/app12168342

https://doi.org/10.3390/separations9120399

https://doi.org/10.1111/ijfs.16359

https://doi.org/10.1111/ijfs.16443 

https://doi.org/10.1111/ijfs.16669

Comments on the Quality of English Language

moderate changes required

Author Response

We want to thank the reviewer for his/her comments. We think that the manuscript is now more complete and comprehensible for the potential reader based on the comments of the three reviewers. In the responses to the reviewers, we have tried to harmonize the comments of all of them.

Dear Author

The review article “New mediators in the crosstalk between different adipose tissues”. The topic is interesting, sounds well to readers, and is suitable for the journal.

Well, the manuscript is good, and the author, please incorporates my suggestion to enhance the effectiveness of this manuscript for the esteemed journal and scientific world. All comments are shared below:

Comment 1: Kindly look about the Grammer mistake in full manuscript.

As the referee suggests we have checked the grammar throughout the manuscript.

Comment 2: Author adds few lines regarding peoples have issues or problem from adipose tissue.

We have added the suggested information in the Introduction section (Page 2, lines 59-65).

Comment 3: Add the table regarding the in vitro and in vivo studies related to adipose tissue.

At the request of the reviewer, we have added a Table (Table 1, pages 9-10) summarizing the main anti-obesity mediators involved in the studies described relating to adipokines, myokines, miRNAs, ... as well as their most important metabolic effects.

Comment 4: Kindly strong the mechanism part of anti-obesity mediators secreted by other tissues with effect in BAT and beige cells.

As the referee suggests we have further described the mechanisms involved in the antiobesity mediators’ actions (Pages 8-9, lines 315-358, 366-369, 374-393).

Comment 5: Author also mentions about herbal approach to cure adiposity or obesity

We have also included data regarding the herbal approach (Page 14, lines 552-556).

Comment 6: Kindly rewrite the conclusion part

We have modified the Conclusions sections to clarify the reader’s understanding.

Comment 7: Author can use these papers to justify the herbal part and mechanism part regarding obesity.

As the reviewer suggests, we have provided information on the mechanisms of some of the natural and herbal products. The bibliographic citations provided by the reviewer have been incorporated into the manuscript [151-153], so we believe that they have improved the understanding of this section. Furthermore, for most natural products, information has been expanded on their mechanisms of action and effects related to prevention and therapy in metabolic pathologies such as obesity.

Round 2

Reviewer 2 Report

Comments and Suggestions for Authors

The introduction is too long, please cut out common sense or unnecessary parts.

In introducing novel adipokines, the article does not provide sufficient background information to explain why these mediators play an important role in the development of obesity, and the authors should have provided more background information, including important findings and theoretical foundations of related studies. And, the authors should cite relevant experimental evidence and data to support the role of these mediators.

The article mentions a number of innovative treatment approaches, but does not provide sufficient detail to explain how these approaches have been applied in experimental models and clinical trials, nor does it provide experimental results for these approaches or for clinical trials.

Comments on the Quality of English Language

Fine.  

Author Response

The introduction is too long, please cut out common sense or unnecessary parts.

As the referee suggests we have reduced the introduction section. We have highlighted the changes in red.

In introducing novel adipokines, the article does not provide sufficient background information to explain why these mediators play an important role in the development of obesity, and the authors should have provided more background information, including important findings and theoretical foundations of related studies. And, the authors should cite relevant experimental evidence and data to support the role of these mediators.

We have introduced the changes suggested by the referee with the corresponding references (lines 157-162, 165-168, 173-187).

The article mentions a number of innovative treatment approaches, but does not provide sufficient detail to explain how these approaches have been applied in experimental models and clinical trials, nor does it provide experimental results for these approaches or for clinical trials.

We have included the details suggested by the referee and a new reference (lines 507-510, 534, 538-544, 557-558, 569-573, 671-673, 687-691, 705-706, 724-725, 730-738).

Reviewer 3 Report

Comments and Suggestions for Authors

Author Add all the comments 

Now we can accept the article in current form

Author Response

Thanks